# Experimental and Clinical Evidence of the Effectiveness of Riboflavin on Migraines

**DOI:** 10.3390/nu13082612

**Published:** 2021-07-29

**Authors:** Gaku Yamanaka, Shinji Suzuki, Natsumi Morishita, Mika Takeshita, Kanako Kanou, Tomoko Takamatsu, Shinichiro Morichi, Yu Ishida, Yusuke Watanabe, Soken Go, Shingo Oana, Hisashi Kawashima

**Affiliations:** Department of Pediatrics and Adolescent Medicine, Tokyo Medical University, Tokyo 160-8402, Japan; shinji-s@tokyo-med.ac.jp (S.S.); sunflowernk69@gmail.com (N.M.); jerryfish_mika@yahoo.co.jp (M.T.); kanako.hayashi.0110@gmail.com (K.K.); t-mori@tokyo-med.ac.jp (T.T.); s.morichi@gmail.com (S.M.); ishiyu@tokyo-med.ac.jp (Y.I.); vandersar_0301@yahoo.co.jp (Y.W.); soupei59@gmail.com (S.G.); oanas@tokyo-med.ac.jp (S.O.); hisashi@tokyo-med.ac.jp (H.K.)

**Keywords:** migraine, riboflavin, mitochondria, inflammation, oxidative stress

## Abstract

Riboflavin, a water-soluble member of the B-vitamin family, plays a vital role in producing energy in mitochondria and reducing inflammation and oxidative stress. Migraine pathogenesis includes neuroinflammation, oxidative stress, and mitochondrial dysfunction. Therefore, riboflavin is increasingly being recognized for its preventive effects on migraines. However, there is no concrete evidence supporting its use because the link between riboflavin and migraines and the underlying mechanisms remains obscure. This review explored the current experimental and clinical evidence of conditions involved in migraine pathogenesis and discussed the role of riboflavin in inhibiting these conditions. Experimental research has demonstrated elevated levels of various oxidative stress markers and pro-inflammatory cytokines in migraines, and riboflavin’s role in reducing these marker levels. Furthermore, clinical research in migraineurs showed increased marker levels and observed riboflavin’s effectiveness in reducing migraines. These findings suggest that inflammation and oxidative stress are associated with migraine pathogenesis, and riboflavin may have neuroprotective effects through its clinically useful anti-inflammatory and anti-oxidative stress properties. Riboflavin’s safety and efficacy suggests its usefulness in migraine prophylaxis; however, insufficient evidence necessitates further study.

## 1. Introduction

Riboflavin (vitamin B2) is an essential water-soluble vitamin that helps prevent various medical conditions, such as sepsis, ischemia, and some cancers [1]. Riboflavin’s biological effects, including antioxidant, anti-aging, anti-inflammatory, and anti-nociceptive effects, have been extensively studied. The pathophysiology of migraines is linked to oxidative stress with mitochondrial dysfunction [2,3,4], and neuroinflammation by the glial cell network [5]. Riboflavin might help improve migraines through various mechanisms, including oxidative stress and neuroinflammation reduction [6].

Riboflavin is heat stable, and cooking does not lower riboflavin levels; however, exposure to light can destroy it. Riboflavin is found in a variety of food sources. Milk products are a rich source, and green vegetables, such as broccoli, collard greens, and turnips, are moderate sources of riboflavin. Surprisingly, 10–15% of the world’s population is genetically restricted in riboflavin absorption and utilization, and there is a potential for biochemical riboflavin deficiency worldwide [7]. Riboflavin deficiency across European countries ranges from 7–20% [8]. Metabolic triggers of migraines, such as fasting and skipping meals, directly link with energy homeostasis and may be associated with riboflavin deficiency. However, there is no evidence that riboflavin deficiency causes or aggravates migraine headaches.

Why is riboflavin administered to migraineurs i.e., patients with migraines? Patients with mitochondrial encephalomyopathy suffer from migraine-like headaches that are relieved by riboflavin; thus, prophylactic riboflavin administration has been attempted [9]. Riboflavin prophylaxis is recommended in adult guidelines [10,11] and has been shown to be somewhat effective in children [12,13]. A migraine is a common yet highly disruptive disease [14]. A rigorous trial on the effectiveness of pharmacological interventions for preventing migraines in children and adolescents found amitriptyline and topiramate to be ineffective [15]. Such results may have led many clinicians to use nutritional supplements with fewer side effects, such as riboflavin, as an optional treatment before using drugs [13,16]. Riboflavin continues to be used for migraine prophylaxis; however, the underlying mechanism of action is still unclear.

In order to better understand the relationship between migraines and riboflavin, this review focuses on the antioxidant and anti-inflammatory properties of riboflavin and mitochondrial damage. In addition, we summarize the current clinical evidence for riboflavin’s efficacy on migraines.

## 2. Search Strategy

This narrative literature review included studies that examined the involvement of oxidative stress, neuroinflammation, and mitochondrial dysfunction in migraines, and riboflavin’s effectiveness as a prophylactic treatment for migraines. A literature search was conducted in the PubMed database and included articles published up to May 2021. Observational studies, randomized controlled trials (RCTs), systematic reviews, and meta-analyses were included, while individual case reports were excluded. The search keywords used were “migraine” and “riboflavin” and “vitamin B2” or “oxidative stress” or “Neuroinflammation” or “inflammation” “cytokine” or “mitochondria”.

## 3. What Is Riboflavin?

Riboflavin was first documented in the late 1870s as a water-soluble yellow pigment found in milk [17]. The pigment was called lactochrome, which was structurally established as riboflavin by the 1930s. Riboflavin is the precursor of metabolically active flavocoenzymes that are utilized as cofactors for approximately 90 flavoproteins in numerous enzymatic reactions, such as flavin adenine dinucleotide (FAD; 84%) or flavin mononucleotide (FMN; 16%) [18]. These flavoproteins (enzymes using flavocoenzymes), including dehydrogenases, oxidases, monooxygenases, and reductases, play critical roles in the mitochondrial electron transport chain, mitochondrial and peroxisomal β-oxidation of fatty acids, citric acid cycle, redox homeostasis, nitric oxide synthases, and branched-chain amino acid catabolism [19]. They are also involved in chromatin remodeling, DNA repair, protein folding, apoptosis, biosynthesis, regulation of other essential cofactors and hormones (including coenzyme A, coenzyme Q, heme, pyridoxal 5′-phosphate, steroids, and thyroxine), and metabolism of other B vitamins and P450 enzymes [7,18].

## 4. Oxidative Stress in Migraines

### 4.1. Clinical Evidence of Oxidative Stress

Studies have indicated the involvement of oxidative stress in migraine pathogenesis and investigated various oxidative stress markers [20,21,22,23]. This review summarizes some of the representative oxidative stress markers and the antioxidant properties of riboflavin (Table 1).

#### 4.1.1. Antioxidants, Total Antioxidant Capacity, and Total Oxidants

A study of patients with migraines without aura showed decreased levels of total antioxidants (TAS), increased levels of total oxidants (TOS), and the oxidative stress index (OSI) compared with controls [23].

Another study found significantly reduced total antioxidant capacity (TAC) in migraineurs compared to controls [24]. However, TAC levels increased after improvement with prophylactic treatment compared to the baseline, and the increase was correlated with treatment success [24]. Furthermore, a recent study confirmed that approximately 40% of recurrent migraineurs had abnormally low TAC [25]. Patients with higher TACs may have less severe migraines, less recent exposure to oxidative stress, and a longer gap between migraine attacks; however, these findings need to be verified [25].

Some studies found no significant differences in TAS, TOS, and OSI levels between migraineurs and controls [26,27]. These conflicting results could be due to differences in the techniques used, biological samples analyzed, timing of sampling, and subject selection [27]. Smokers have higher levels of malondialdehyde (MDA), 8-hydroxy-2-deoxyguanosine (8-OHdG), superoxide dismutase, and glutathione peroxidase [28]. Women have lower oxidative stress levels than men [29]. Furthermore, nonsteroidal anti-inflammatory drugs and paracetamol may be directly or indirectly involved in oxidative and antioxidative processes in the body [30].

#### 4.1.2. Peroxide and Malondialdehyde

Lipid peroxidation refers to the oxidative degradation of lipids, especially polyunsaturated fatty acids, by free radicals in cell membranes. MDA is the end product of lipid peroxidation. Increased MDA [22,31,32,33,34] and peroxide [25] levels have been found in migraineurs, especially in cases of migraines with aura.

#### 4.1.3. 8-Hydroxy-2-deoxyguanosine

Plasma 8-OHdG is an indicator of oxidative DNA damage. A study matched variables in both groups to minimize the effects of smoking, sex, and use of symptomatic medication and detected no significant differences in TOS or TAS, but significantly higher plasma 8-OHdG levels in migraineurs, especially in cases of migraine without aura [27]. The higher 8-OHdG levels, related to more severe and frequent attacks [27], might be a more sensitive oxidative stress marker for migraines.

#### 4.1.4. Alpha-Lipoic Acid

Recent research has shown that patients with high-frequency episodic migraines (5 to 14 migraine days per month) showed low values of alpha-lipoic acid (ALA or thioctic acid) [25], which functions as a water and fat-soluble antioxidant [35,36]. ALA plays a vital role as a coenzyme in energy [35,36]. Interestingly, ALA intake also significantly reduces the frequency, severity, and duration of migraine attacks [37,38].

**Table 1 nutrients-13-02612-t001:** Summary of oxidative stress markers in migraineurs.

Markers	Design	Sample Size	Findings	Reference
Antioxidants (TAS), total antioxidant capacity (TAC), and total oxidants (TOS)	Case-control	75/65	Patients’ serum TAS levels were significantly lower than those of healthy controls. Serum TOS values were significantly higher in patients than in control. The mean values of oxidative stress index (OSI) were greater in patients than in controls.	Alp et al., 2010 [23]
	Case-control	141/70	TAS, TOS, OSI had no statistical difference between the patients and controls.	Eren et al., 2015 [26]
	Case-control	50/30	No significantly different values of TAS, TOS, and OSI found in migraineurs.	Geyik et al., 2016 [27]
	Before and after	120/30	TAC levels were increased following transcranial magnetic stimulation and amitriptyline.	Tripathi et al., 2018 [24]
	Case series	32/14	Decreased serum TAC levels found in 37.5% of patients.	Gross et al., 2021 [25]
Peroxide and malondialdehyde (MDA)	Case-control	39/30	While migraine with/without aura patients had low platelet superoxide dismutase (SOD) concentrations, platelet SOD activity decreased only in migraine with aura patients.	Shimomura et al., 1994 [31]
	Case-control	56/25	The MDA levels of migraineurs were significantly higher than controls. The SOD activity was significantly higher in the migraine with aura than migraine without aura. No significant correlation was found between these levels and headache attack duration.	Tuncel et al., 2008 [22]
	Case-control	50/50	Migraineurs had significantly high MDA and “ferric reducing ability of plasma” levels compared to the other two groups (tension-type headache and control group).	Gupta et al., 2009 [32]
	Case-control	48/48	There was no significant difference in MDA concentration between migraineur and control groups. Significantly increased 4-hydroxynonenal levels were found in the migraine group compared to the control group.	Bernecker et al., 2011 [34]
	Case-control	32/14	In the migraine group, catalase was significantly lower and MDA concentrations were higher than controls. Serum catalase levels were significantly lower in migraineurs with deep white matter hyperintensities than in migraineurs without deep white matter hyperintensities and in controls	Aytaç et al., 2014 [33]
	Case series	32	High serum peroxide levels were found in 46.9% of patients.	Gross et al., 2021 [25]
8-hydroxy-2-deoxyguanosine (8-OHdG)	Case-control	50/30	Increased plasma 8-OHdG levels were shown in migraineurs.	Geyik et al., 2016 [27]
Alpha-lipoic acid (ALA)	Randomized controlled trial	44	In a within-group analysis, patients who received thioctic acid (ALA) for three months had a significant reduction in the frequency of attacks, number of headache days, and severity of headaches, while these outcomes remained unchanged in the placebo group. The proportion of 50% responders was not significantly different between thioctic acid (30.8%) and the placebo (27.8%).	Magis et al., 2007 [37]
	Before and after	32	The percentage of patients with a 50% or greater reduction in attacks was significantly reduced at 2, 4, and 6 months. The incidence rate ratio of attacks at 6 months was significantly decreased compared to the baseline.	Cavestro et al., 2018 [38]
	Case series	32	Decreased serum ALA levels were found in 87.5% of patients.	Gross et al., 2021 [25]

### 4.2. Experimental Evidence of Oxidative Stress

Cortical spreading depression (CSD), a characteristic feature of migraines with aura, represents a strong wave of neuronal depolarization associated with the activation of glial cells and blood vessels [39,40].

Dramatic metabolic changes in the cerebral cortex associated with intracellular calcium overload during CSD could induce transient oxidative stress [41,42]. CSD causes oxidative stress in the cerebral cortex, meninges, and even in the trigeminal ganglion, which is not directly exposed to the trigger substance [43]. Additionally, it suggests a direct stimulatory effect of reactive oxygen species (ROS) on nociceptor firing via transient receptor potential ankyrin subtype 1 (TRPA1) ion channels and an indirect role for ROS in sensitizing sensory afferents via the release of a major migraine mediator calcitonin gene-related peptide (CGRP) from nociceptor neurons [43]. TRPA1 ion channels enable CGRP release from dural afferents, and mediate the behavioral picture of neurogenic inflammation and migraines in animal models [44]. TRPA1 undergoes oxidative stress and initiates a neuroinflammatory response in migraines. In other words, TRPA1 might be a bridge between oxidative stress and neuroinflammation in migraines [44].

### 4.3. Riboflavin as an Antioxidative Agent

Multiple animal studies have confirmed that lipid peroxidation marker levels are elevated when riboflavin is deficient, and the levels reduce after administering riboflavin [45,46,47]. Riboflavin’s antioxidant function is possibly due to its action on glutathione’s redox cycle, the reductive oxidation reaction of riboflavin itself, and antioxidant enzyme activity.

Glyceryl trinitrate (GTN) infusion is a reliable method to induce migraine-like allodynia and sensitization in experimental animals. In the GTN-induced model’s analysis, the brain and microsomal lipid peroxidation levels were higher in the GTN-induced models than in healthy rats [48,49]. Additionally, co-administration of riboflavin and vitamin E [48] or selenium [49], the essential antioxidant trace elements, showed a protective effect on GTN-induced brain injury by inhibiting free radical production.

## 5. Neuroinflammation

Neurogenic inflammation, a major factor in migraine pathogenesis, is characterized by the release of neuropeptides (e.g., substance P and CGRP) from the trigeminal nerve, leading to arterial vasodilation, plasma protein extravasation, and mast cell degranulation. The involvement of these neuropeptides in migraines is evident, and several inflammatory markers have been identified in migraineurs [50,51].

### 5.1. Clinical Evidence of Neuroinflammation

Cytokine expression increases in the CSF and plasma of both adult [52,53,54,55,56] and pediatric [57] migraineurs. Interleukin (IL)-1β, IL-6, IL-10, and tumor necrosis factor (TNF)-α levels are elevated in the blood during or early in the migraine pain period compared to the non-migraine pain period [52,53,54]. Interestingly, elevated serum levels of ICAM1, a marker of vascular endothelial damage, have also been reported in migraines with aura compared to those with migraines without aura [56]. Furthermore, despite normal serum TNFα levels in patients with chronic migraines, elevated TNFα levels in the CSF have been identified. Persistently elevated TNFα levels in the CSF may be one of the causes of treatment-resistant headaches, which does not improve despite aggressive admission interventions [55]. Other inflammatory markers associated with vascular diseases, such as Hcy and matrix metalloproteinase-9 (MMP-9), are elevated in migraineurs’ blood [58], indicating neuroinflammatory cascade activation in migraine pathogenesis.

### 5.2. Experimental Evidence of Neuroinflammation

CSD triggers a substantial inflammatory response via the opening of the neuronal Pannexin1 mega-channels, activation of caspase-1, followed by the release of high mobility group box 1 (HMGB1) from the neurons, and activation of NF-κB in astrocytes [59]. Furthermore, the development of CSD causes inflammation of the meninges with activation of macrophages and mast cells and increased cytokine levels [60,61]. CSD can also activate astrocytes [62,63] and enhance the production of several inflammatory cytokines, such as IL-1b, IL-6, and TNFα [5,64,65], as well as toll-like receptors (TLR3 and TLR4) in cultured astrocytes [5]. These results demonstrate the existence of a neuroinflammatory process through CSD, one of the hallmarks of migraines with aura.

### 5.3. Riboflavin as an Anti-Inflammatory Agent

Riboflavin can inhibit trypsin-like proteasome activity and suppress NF-κB activation, as well as the resulting production of TNFα and nitric oxide (NO) in experimental models by LPS [66], and by a staphylococcal infectious model [67]. NF-κB regulates the acute phase of the inflammatory response, and is also involved in the painful state of the central nervous system caused by chronic inflammation; NF-κB inhibition in astroglia reduces the painful state [68]. Valproic acid, a migraine prophylactic, inhibits NF-κB activation in the trigeminocervical complex [69] and alleviates nitroglycerin-induced migraines [69].

In staphylococcal infections, riboflavin exerts an anti-inflammatory effect by reducing NF-κB synthesis, leading to decreased NO and TNFα levels, and modulates the rise of the anti-inflammatory cytokine IL-10 and the function of MCP-1 (monocyte chemoattractant protein 1), a potent chemoattractant [67]. In the sepsis-associated multiple organ failure, riboflavin inhibits the release and expression of HMGB1 [70], which is produced by neurons during the pathogenesis of experimental migraines [59].

Thus, riboflavin has been shown to exert anti-inflammatory effects under pathological conditions where inflammation is present. Also, the inflammatory profile, mainly NF-κB, plays an important role in the pathogenesis of a migraine, and NF-κB suppression by riboflavin might contribute to treating migraines by regulating inflammatory cytokines and HMGB1.

## 6. Mitochondrial Dysfunction

Mitochondria play an important role in a wide range of cellular functions, such as energy generation, ROS production, Ca^2+^ homeostasis regulation, and apoptosis [71]. Mitochondrial disease symptoms occur in almost all organs, but primarily in high energy-consuming organs, such as the brain and muscles [71].

### 6.1. Clinical Evidence of Mitochondrial Dysfunction

Although structural abnormalities of mitochondria in diseases such as mitochondrial encephalomyopathy are well known to affect mainly non-active mitotic tissues such as skeletal muscle and neurons [72,73], ragged red fibers and cytochrome-c-oxidase (COX)-negative fibers have also been found in the skeletal muscles of patients with migraines with prolonged aura [2,74]. The electron microscope also confirms that migraineurs have large mitochondrial clusters containing sub-crystals [2]. In addition, magnetic resonance spectroscopy studies have revealed reduced interictal rates of mitochondrial oxidative phosphorylation in migraineurs’ brains and muscles [75,76,77].

Interestingly, the seemingly unrelated migraine triggers, such as ovarian hormone changes, weather changes, alcohol, strong smells, strong light, and loud noises, have a potential common denominator in the form of changes in the mitochondrial metabolism and oxidative stress [78,79]. Disturbances in mitochondrial metabolism might contribute to the pathogenesis of migraines by lowering the threshold for migraine attack propagation [80,81]. Furthermore, mitochondrial genome analysis demonstrated that polymorphisms account for a significant portion of the genetic factors involved in migraine etiology [82], and clinical evidence of the link between migraine and mitochondrial dysfunction is slowly accumulating [2,75,76,83,84].

### 6.2. Experimental Evidence of Mitochondrial Dysfunction

This current research showed that common migraine triggers have the ability to generate oxidative stress through mitochondrial dysfunction, calcium excitotoxicity, microglia and NADPH oxidase activation, and as a byproduct of monoamine oxidase (MAO), cytochrome P450, or NO synthase [78]. In particular, mitochondria are key to the primary mechanism of intracellular Ca^2+^ sequestration; therefore, mitochondrial dysfunction can lead to pain hypersensitivity [85]. Vasoconstriction during CSD is also triggered by an increase in Ca^2+^ concentrations in the astrocytes through a process mediated by phospholipase A2, a metabolite of arachidonic acid [86]. Mitochondria play a crucial role in the normal functioning of neurons, and a Ca^2+^ imbalance can lead to an imbalance in various downstream processes, and thus further increase susceptibility to migraines [3].

The migraine model demonstrated abnormalities in the mitochondrial biogenesis capacity of trigeminal neurons, with reduced copy numbers of mitochondrial DNA and altered mRNA levels of the peroxisome proliferator-activated receptor-γ coactivator 1-α [87], which are essential regulators of mitochondrial biogenesis [23]. These experimental findings indicate that mitochondrial dysfunction is an important hallmark of migraines.

Mechanistically, riboflavin prevented mitochondrial perturbations, such as mitochondrial ROS production and mitochondrial DNA release, which trigger NLRP3 inflammasome assembly. Furthermore, riboflavin disrupts the caspase-1 activity and inhibits AIM2, NLRC4, and non-canonical inflammasomes. Therefore, riboflavin has antioxidant and anti-inflammasome properties that regulate the inflammatory response [88]. These results collectively suggest that oxidative stress and neuroinflammatory responses due to mitochondrial dysfunction influence the pathogenesis of migraine, and that these mechanisms might be suppressed by riboflavin.

## 7. Clinical Evidence of Riboflavin Efficacy

We summarized the clinical studies of riboflavin, distinguishing between children, adolescents (Table 2), and adults (Table 3).

Seven previous studies, including four RCTs, evaluated riboflavin’s role in preventing pediatric migraines (Table 1). All studies except for two RCTs reported that riboflavin was effective [89,90,91,92]; the other two studies lacked evidence to support the use of riboflavin for pediatric migraines [93,94]. The following factors reportedly inhibit the effects of riboflavin: the presence of comorbid headaches, male sex, and age under 12 years [91,95].

Seven studies (including three RCTs) in adults have evaluated the role of riboflavin in preventing adult migraines (Table 2). The dose of riboflavin was 400 mg, except in one case (100 mg) [11]. All studies demonstrated the effectiveness of riboflavin [9,11,96,97,98,99,100].

**Table 2 nutrients-13-02612-t002:** Summary of studies on riboflavin for pediatric migraines.

Study Design	N	Intervention	Comparison	Outcomes	Reference
RCT	48	Riboflavin (200 mg daily) for 12 weeks (*n* = 27)	Placebo for 12 weeks (*n* = 21)	No difference between the comparison groups in terms of the proportion of participants with 50% or greater reduction in migraine frequency (*p* = 0.125)	* MacLennan et al., 2008 [93]
Before-after study	41	Riboflavin (200 mg or 400 mg daily) for three, four, or six months	Baseline period	Significant reduction in headache frequency after treatment for three or four months (*p* < 0.01), which was not sustained at six months (*p* > 0.05)	Condo et al., 2009 [95]
Crossover RCT	42	Riboflavin (50 mg daily) for four months (*n* = 20)	Placebo for four months(*n* = 22)	No difference between the comparison groups in terms of change in migraine frequency (*p* = 0.44); the riboflavin group showed a greater reduction in the frequency of tension-type headaches than the placebo group (*p* = 0.04)	* Bruijn et al., 2010 [94]
RCT	98	Riboflavin (400 mg daily) for three months (*n* = 50)	Placebo for three months (*n* = 48)	Headache frequency decreased from the first month to the second month, and to the third month (3.7 per month); headache duration also decreased (*p*-values: 0.012 and 0.001, respectively) compared to the placebo group. Disability, as measured by the PedMIDAS, also decreased (*p* = 0.001).	Athaillah et al., 2012 [89]
RCT	90	Riboflavin (200 mg or 400 mg daily) for 3 months (*n* = 30, and 30, respectively)	Placebo for three months (*n* = 30)	The riboflavin 400 mg group showed a greater reduction in the headache frequency and duration than the placebo (*p* = 0.00 for both).	Talebian et al., 2018 [90]
Retrospective observational study	68	Riboflavin (10 or 40 mg daily) for three months (*n* = 13 and 55, respectively)	N/A	Significant overall reduction detected in the median frequency of headache episodes from baseline to three months (*p* = 0.00).	Yamanaka et al., 2020 [91]
Retrospective observational study	42	Riboflavin (100 and 200 mg for children weighing 20 to 40 kg and greater than 40 kg, respectively)	N/A	Significant decrease in the frequency of headache days after 2–4 months compared to the baseline. Mean headache intensity (*p* < 0.001), and headache duration (*p* < 0.001) decreased significantly.	Das et al., 2020 [92]

RCT, randomized controlled trial; MIDAS, Migraine Disability Assessment; N/A, not available. The asterisk (*) indicates a study with a negative result.

**Table 3 nutrients-13-02612-t003:** Summary of studies on riboflavin for adult migraine.

Study Design	N	Intervention	Comparison	Outcomes	Reference
Open label trial	44	Riboflavin 400 mg daily (23/44 received aspirin 75 mg daily)	N/A	A 68.2% improvement in migraine severity score, no difference between the aspirin-treated and non-aspirin-treated groups (*p*-value not reported)	Schoenen et al., 1994 [96]
RCT	55	400 mg daily	Placebo for three months (*n* = 27)	Riboflavin significantly reduced the frequency of seizures (*p* = 0.005) and the number of headache days (*p* = 0.012) when compared with the placebo group	Schoenen et al., 1998 [9]
Open label trial	26	400 mg daily vs. bisoprolol 10 mg daily or metoprolol 200 mg daily	N/A	Headache frequency was significantly reduced (*p* < 0.05) in both groups, but there was no difference between the two groups	Sándor et al., 2000 [97]
Open label trial	23	400 mg daily	N/A	Headache frequency significantly decreased from 4 days/month at baseline to 2 days/month at three and six months (*p* < 0.05)	Boehnke et al., 2004 [98]
Open label trial	64	400 mg daily	N/A	62.5% responded and haplotype H was associated with a reduced probability of responding to riboflavin (OR, 0.24; 95% confidence interval [0.08, 0.71])	Di Lorenzo et al., 2009 [99]
RCT	100	100 mg daily for at least three months	Propranolol 80 mg daily for at least three months (*n* = 50)	A greater reduction in migraine frequency in the propranolol group at one month (*p* < 0.001), but no difference between the groups at three and six months	Nambiar NJ et al., 2011 [11]
RCT	90	400 mg/day	Sodium valproate 500 mg/day	The frequency, median duration per month, and severity of headache decreased in both groups, but the difference between them was not significant (*p* > 0.05). However, the vitamin B2 group had significantly fewer side effects (*p* = 0.005).	Rahimd et al., 2015 [100]

RCT, randomized controlled trial; N/A, not available.

A small number of participants (*n* = 48 and 42) and a low dosage of riboflavin (25 mg or 100 mg) might have contributed to the negative results reported by both RCTs in pediatric patients. Although higher doses are more effective [90], pharmacological action of even small doses of riboflavin (25 mg or 100 mg) have been reported [91]. Even an adult migraine study indicated the effects of low-dose (100 mg) supplementation [11]. The exact riboflavin requirements, even in adults with a migraine, is unclear; a dose of 400 mg was chosen in a previous adult study because similar high doses were previously used to treat mitochondrial disorders and because of riboflavin’s lack of toxicity [9]. A pharmacokinetic study in adults did not find significant differences in blood concentrations, maximal serum concentrations, or areas under the curve with 20, 40, and 60-mg riboflavin supplementations [101]. Our recent univariate analysis also showed no difference between 10 mg and 40 mg doses [91]. Therefore, lower doses may exert pharmacological effects in children.

Although no serious side effects have been reported to date, and there is a widespread perception that riboflavin has no side effects, some adverse effects have been reported. Studies utilizing high doses of riboflavin reported orange discoloration of the urine, polyuria, diarrhea, vomiting, and an increased appetite without weight gain [9,89,93,95]; studies using low doses did not report any adverse effects [91,94].

Prophylactic administration of riboflavin is recommended in the guidelines for adults because of its cost, usefulness [10,11], and lack of serious adverse effects. Moreover, both adult and pediatric studies have shown favorable outcomes. However, there is no solid evidence owing to the low-quality evidence and the limited number of studies. Larger studies with longer observational periods are required to assess the effectiveness of riboflavin-based prophylaxis for migraines. We believe that riboflavin could be used as an option to prevent migraines.

## 8. Conclusions

In this review, we present the current clinical and experimental evidence of inflammation, oxidative stress, and mitochondrial dysfunction playing a role in migraine pathogenesis. We addressed the possibility of inhibiting these mechanisms with riboflavin, and demonstrated its clinical benefits in preventing migraine headaches.

Research in migraineurs has demonstrated elevated levels of various oxidative stress markers and pro-inflammatory cytokines, and a migraine preventive effect of riboflavin. Animal models have also shown that riboflavin reduces these markers. These findings suggest that inflammation and oxidative stress are associated with the pathogenesis of migraines, and that riboflavin may have neuroprotective effects through its clinically useful anti-inflammatory and antioxidant stress properties.

Although the inflammatory response associated with CSD has been demonstrated in experimental mice, there is a lack of validation of the inflammatory response, oxidative stress, and mitochondrial protection in other mouse models of migraines. Also, further validation of the neuroprotective effects of riboflavin is required.

Clinical studies would need to go beyond simply confirming the effect of riboflavin administration on migraines, and distinguish between riboflavin-deficient and non-riboflavin-deficient cases, while simultaneously examining a range of inflammatory and oxidative stress markers.

Although there is currently no strong clinical evidence to support the use of riboflavin, the findings from the existing studies are encouraging. Larger RCTs are warranted, but are difficult to implement because riboflavin is inexpensive and of no commercial interest to the pharmaceutical industry. Considering its efficacy and safety, riboflavin may be used for migraine prophylaxis.

## Data Availability

The datasets generated and/or analyzed during the current study are available at the PubMed database repository (https://pubmed.ncbi.nlm.nih.gov/, accessed on 29 July 2021).

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
