# Peer review of "Experimental and Clinical Evidence of the Effectiveness of Riboflavin on Migraines"

_nutrients, 2021, doi:10.3390/nu13082612_

Round 1

Reviewer 1 Report

This is a very useful review on the biological effects of riboflavin in relation to migraine pathogenesis and on published studies of riboflavin's efficacy in migraine prevention.

Concerning the discrepancy of results in clinical trials of riboflavin in migraine, besides the difference in dosages, the authors might mention differences in 1) pharmaceutical quality and conditioning of the product , 2) timing of administration (riboflavin's GI absorption is limited, but it is thought to be absorbed in several waves over 24 hours; several substances like magnesium compete with absorption in the intestine...)

I have comments and suggestions on some questionable statements.

  • line 52: "no established prophylactic treatment exist" is doubtless an erroneous statement and probably not what the authors intended to write (see ref 10 for proof of the opposite!!)
  • lines 54-55: "amitriptyline and topiramate...widely used prophylactic treatments for migraines in children" These drugs are not widely used in children because of their side effects. Ref 15 is only one study, which by the way is negative.
  • line 98: "after improved prophylactic treatment.." likely should read "after improvement with prophylactic..."
  • line 142: "sensitizing the senses" what do you mean: ? "sensitizing sensory afferents" ?
  • line 144: "activated sural afferents" I doubt that the sural nerve is implicated in animal models of migraine! Should be "dural afferents".
  • line 156: "rats with migraine headaches" GTN induces allodynia and sensitization in rats, not migraine headaches
  • line 188: "CSD, one of the hallmarks of migraine" should read "one of the hallmarks of migraine with aura"
  • line 202 "..during migraine pathogenesis": again there is confusion between experimental models of migraine and clinical migraine; HMGB1 is expressed after experimental CSD in rats and has not (yet) been demonstrated in migraineurs.
  • line 216: "lagged red fibers" should read "ragged red fibers"
  • line 307: "have also showed" = "have also shown"
  • line 319: "there is currently little evidence" is an understatement that does not correspond to the data shown and the rationale of the review; more correct would be the following: "there is currently no strong clinical evidence to ... encouraging. Larger RCTs are warranted, but difficult to implement because Riboflavin is inexpensive and of no commercial interest to the pharmaceutical industry. "

Author Response

Manuscript ID: nutrients-1285874

Type of manuscript: Review

Title: Experimental and Clinical Evidence of the Effectiveness of Riboflavin in Migraine

To the Reviewer 1

We appreciate the time and effort that the reviewer has dedicated to providing insightful feedback. We have revised our manuscript according to your comments as much as possible. All revisions and additions are indicated in underlined, yellow text in the manuscript. We sincerely hope that with these revisions our manuscript will be suitable for publication in Nutrients.

Referee 1

Comments and suggestions.

  1. line 52: "no established prophylactic treatment exist" is doubtless an erroneous statement and probably not what the authors intended to write (see ref 10 for proof of the opposite!!

Response. As you pointed out, there are some drugs recommended by the American Academy of Neurology and the American Headache Society, so we have omitted the following; however, no established prophylactic treatment exists, especially for children and adolescents.

  1. lines 54-55: "amitriptyline and topiramate...widely used prophylactic treatments for migraines in children" These drugs are not widely used in children because of their side effects. Ref 15 is only one study, which by the way is negative.

Response.  Thank you for your comment. I've edited it below; the widely used prophylactic treatments for migraines in children,

  1. line 98: "after improved prophylactic treatment.." likely should read "after improvement with prophylactic..."

Response. we changed as you suggested.

  1. line 142: "sensitizing the senses" what do you mean: ? "sensitizing sensory afferents" ?    Thank you for pointing out the error. I have corrected it.

  1. line 144: "activated sural afferents" I doubt that the sural nerve is implicated in animal models of migraine! Should be "dural afferents".

Response. I suppose you're absolutely right. We changed as you suggested.

  1. line 156: "rats with migraine headaches" GTN induces allodynia and sensitization in rats, not migraine headaches

Response. As you mentioned, we changed it as follows.

Line 152,

Before, Glyceryl trinitrate (GTN) infusion is a reliable method to induce migraine-like headaches in experimental animals.

 to

Glyceryl trinitrate (GTN) infusion is a reliable method to induce migraine-like allodynia and sensitization in experimental animals.

Line 152,

in the GTN-induced rats with migraine headaches than in healthy rats

to

 in the GTN-induced models than in healthy rats

  1. line 188: "CSD, one of the hallmarks of migraine" should read "one of the hallmarks of migraine with aura"

Response. We've changed it as you suggested.

  1. line 202 "..during migraine pathogenesis": again there is confusion between experimental models of migraine and clinical migraine; HMGB1 is expressed after experimental CSD in rats and has not (yet) been demonstrated in migraineurs.

Response. The following changes have been made.

“during migraine pathogenesis” to “during the pathogenesis of experimental migraine”

  1. line 216: "lagged red fibers" should read "raggedred fibers"

line 307: "have also showed" = "have also shown"

Response. We've changed it as you suggested.

  1. line 319: "there is currently little evidence" is an understatement that does not correspond to the data shown and the rationale of the review; more correct would be the following: "there is currently no strong clinical evidence to ... encouraging. Larger RCTs are warranted, but difficult to implement because Riboflavin is inexpensive and of no commercial interest to the pharmaceutical industry. "

Response. Finally, we would like to thank you for your valuable comments. We have improved the content as you suggested. Thank you once again.

Reviewer 2 Report

Congrats! It's a well written review about an interesting issue. A review paper about the role of rivoflavin in migraine pathogenesis and in preventive migraine treatments.

A view comments only:

1) Table 1, L131: it would be better to use only the term "migraineurs" and not "migraine patients" (or the oposite) in the title of the table and in the text and not both terms

2) L169-170: "in the seizure period..... to the non-seizure period". Is this right? Do you mean during the migraine pain period?

Author Response

Manuscript ID: nutrients-1285874

Type of manuscript: Review

Title: Experimental and Clinical Evidence of the Effectiveness of Riboflavin in Migraine

To the Reviewer 3

We appreciate the time and effort that the reviewer has dedicated to providing insightful feedback. We have revised our manuscript according to your comments as much as possible. All revisions and additions are indicated in underlined, yellow text in the manuscript. We sincerely hope that with these revisions our manuscript will be suitable for publication in Nutrients.

Reviewer: 2
Comments

  • Table 1, L131: it would be better to use only the term "migraineurs" and not "migraine patients" (or the oposite) in the title of the table and in the text and not both terms

Response, Thank you for your comment. As you pointed out, we have standardized on "migraineurs".

2) L169-170: "in the seizure period..... to the non-seizure period". Is this right? Do you mean during the migraine pain period?

Response, We apologize for the lack of clarity in the text. We have corrected it as follows.

in the migraine pain period compared to the non-migraine pain period